# Eigenoption Discovery through the Deep Successor Representation

**Marlos C. Machado**[1]*, **Clemens Rosenbaum**[2], **Xiaoxiao Guo**[3]
**Miao Liu**[3], **Gerald Tesauro**[3], **Murray Campbell**[3]
[1] University of Alberta, Edmonton, AB, Canada
[2] University of Massachusetts, Amherst, MA, USA
[3] IBM Research, Yorktown Heights, NY, USA

## Abstract

Options in reinforcement learning allow agents to hierarchically decompose a task into subtasks, having the potential to speed up learning and planning. However, autonomously learning effective sets of options is still a major challenge in the field. In this paper we focus on the recently introduced idea of using representation learning methods to guide the option discovery process. Specifically, we look at *eigenoptions*, options obtained from representations that encode diffusive information flow in the environment. We extend the existing algorithms for eigenoption discovery to settings with *stochastic* transitions and in which *handcrafted* features are not available. We propose an algorithm that discovers eigenoptions while learning non-linear state representations from raw pixels. It exploits recent successes in the deep reinforcement learning literature and the equivalence between proto-value functions and the successor representation. We use traditional tabular domains to provide intuition about our approach and Atari 2600 games to demonstrate its potential.

## 1 Introduction

Sequential decision making usually involves planning, acting, and learning about temporally extended courses of actions over different time scales. In the reinforcement learning framework, *options* are a well-known formalization of the notion of actions extended in time; and they have been shown to speed up learning and planning when appropriately defined (*e.g.*, Brunskill & Li, 2014; Guo et al., 2017; Solway et al., 2014). In spite of that, autonomously identifying good options is still an open problem. This problem is known as the problem of option discovery.

Option discovery has received ample attention over many years, with varied solutions being proposed (*e.g.*, Bacon et al., 2017; Şimşek & Barto, 2004; Daniel et al., 2016; Florensa et al., 2017; Konidaris & Barto, 2009; Mankowitz et al., 2016; McGovern & Barto, 2001). Recently, Machado et al. (2017) and Vezhnevets et al. (2017) proposed the idea of learning options that traverse directions of a latent representation of the environment. In this paper we further explore this idea.

More specifically, we focus on the concept of *eigenoptions* (Machado et al., 2017), options learned using a model of diffusive information flow in the environment. They have been shown to improve agents' performance by reducing the expected number of time steps a uniform random policy needs in order to traverse the state space. Eigenoptions are defined in terms of proto-value functions (PVFs; Mahadevan, 2005), basis functions learned from the environment's underlying state-transition graph. PVFs and eigenoptions have been defined and thoroughly evaluated in the tabular case. Currently, eigenoptions can be used in environments where it is infeasible to enumerate states only when a *linear* representation of these states is *known beforehand*.

In this paper we extend the notion of eigenoptions to *stochastic* environments with *non-enumerated* states, which are commonly approximated by feature representations. Despite methods that learn representations generally being more flexible, more scalable, and often leading to better performance, current algorithms for eigenoption discovery cannot be combined with representation learn-

---

*Corresponding author: `machado@ualberta.ca`

ing. We introduce an algorithm that is capable of discovering eigenoptions while learning representations. The learned representations implicitly approximate the model of diffusive information flow (hereafter abbreviated as the DIF model) in the environment. We do so by exploiting the equivalence between PVFs and the successor representation (SR; Dayan, 1993). Notably, by using the SR we also start to be able to deal with stochastic transitions naturally, a limitation of previous algorithms.

We evaluate our algorithm in a tabular domain as well as on Atari 2600 games. We use the tabular domain to provide intuition about our algorithm and to compare it to the algorithms in the literature. Our evaluation in Atari 2600 games provides promising evidence of the applicability of our algorithm in a setting in which a representation of the agent's observation is learned from raw pixels.

## 2 BACKGROUND

In this section we discuss the reinforcement learning setting, the options framework, and the set of options known as eigenoptions. We also discuss the successor representation, which is the main concept used in the proposed algorithm.

### 2.1 REINFORCEMENT LEARNING AND OPTIONS

We consider the reinforcement learning (RL) problem in which a learning agent interacts with an unknown environment in order to maximize a reward signal. RL is often formalized as a Markov decision process (MDP), described as a 5-tuple: $\langle \mathcal{S}, \mathcal{A}, p, r, \gamma \rangle$. At time $t$ the agent is in state $s_t \in \mathcal{S}$ where it takes action $a_t \in \mathcal{A}$ that leads to the next state $s_{t+1} \in \mathcal{S}$ according to the transition probability kernel $p(s'|s, a)$. The agent also observes a reward $R_{t+1}$ generated by the function $r : \mathcal{S} \times \mathcal{A} \to \mathbb{R}$. The agent's goal is to learn a policy $\pi : \mathcal{S} \times \mathcal{A} \to [0, 1]$ that maximizes the expected discounted return $G_t \doteq \mathbb{E}_{\pi,p}\left[ \sum_{k=0}^{\infty} \gamma^k R_{t+k+1} | s_t \right]$, where $\gamma \in [0, 1]$ is the discount factor.

In this paper we are interested in the class of algorithms that determine the agent's policy by being greedy with respect to estimates of value functions; either w.r.t. the state value $v_\pi(s)$, or w.r.t. the state-action value function $q_\pi(s, a)$. Formally, $v_\pi(s) = \mathbb{E}_{\pi,p}[G_t|s] = \sum_a \pi(a|s)q_\pi(s, a)$. Notice that in large problems these estimates have to be approximated because it is infeasible to learn a value for each state-action pair. This is generally done by parameterizing $q_\pi(s, a)$ with a set of weights $\boldsymbol{\theta}$ such that $q(s, a, \boldsymbol{\theta}) \approx q_\pi(s, a)$. Currently, neural networks are the most successful parametrization approach in the field (*e.g.*, Mnih et al., 2015; Tesauro, 1995). One of the better known instantiations of this idea is the algorithm called Deep Q-network (DQN; Mnih et al., 2015), which uses a neural network to estimate state-action value functions from raw pixels.

Options (Sutton et al., 1999) are our main topic of study. They are temporally extended actions that allow us to represent courses of actions. An option $\omega \in \Omega$ is a 3-tuple $\omega = \langle \mathcal{I}_\omega, \pi_\omega, \mathcal{T}_\omega \rangle$ where $\mathcal{I}_\omega \subseteq \mathcal{S}$ denotes the option's initiation set, $\pi_\omega : \mathcal{S} \times \mathcal{A} \to [0, 1]$ denotes the option's policy, and $\mathcal{T}_\omega \subseteq \mathcal{S}$ denotes the option's termination set. We consider the *call-and-return* option execution model in which a meta-policy $\mu : \mathcal{S} \to \Omega$ dictates the agent's behavior (notice $\mathcal{A} \subseteq \Omega$). After the agent decides to follow option $\omega$ from a state in $\mathcal{I}_\omega$, actions are selected according to $\pi_\omega$ until the agent reaches a state in $\mathcal{T}_\omega$. We are interested in learning $\mathcal{I}_\omega, \pi_\omega$, and $\mathcal{T}_\omega$ from scratch.

### 2.2 PROTO-VALUE FUNCTIONS AND EIGENOPTIONS

Eigenoptions are options that maximize eigenpurposes $r_i^{\mathbf{e}}$, intrinsic reward functions obtained from the DIF model (Machado et al., 2017). Formally,

$$r_i^{\mathbf{e}}(s, s') \;\; = \;\; \mathbf{e}^\top \Big( \boldsymbol{\phi}(s') - \boldsymbol{\phi}(s) \Big), \tag{1}$$

where $\phi(\cdot)$ denotes a feature representation of a given state (*e.g.*, one-hot encoding in the tabular case) and $\mathbf{e}$ denotes an eigenvector encoding the DIF model at a specific timescale. Each intrinsic reward function, defined by the eigenvector being used, incentivizes the agent to traverse a different latent dimension of the state space.

In the tabular case, the algorithms capable of learning eigenoptions encode the DIF model through the combinatorial graph Laplacian $\mathcal{L} = D^{-1/2}(D - W)D^{-1/2}$, where $W$ is the graph's weight matrix and $D$ is the diagonal matrix whose entries are the row sums of $W$. The weight matrix

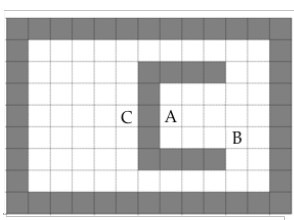 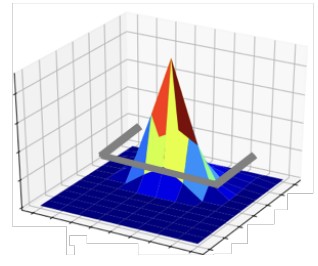 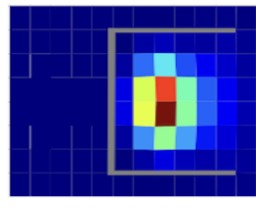

Figure 1: Successor representation, with respect to the uniform random policy, of state A (left). This example is similar to Dayan's (1993). The red color represents larger values while the blue color represents smaller values (states that are temporally further away).

is a square matrix where the $ij$-th entry represents the connection between states $i$ and $j$. Notice that this approach does not naturally deal with stochastic or unidirectional transitions because $W$ is generally defined as a *symmetric* adjacency matrix. Importantly, the eigenvectors of $\mathcal{L}$ are also known as proto-value functions (PVFs; Mahadevan, 2005; Mahadevan & Maggioni, 2007).

In settings in which states cannot be enumerated, the DIF model is represented through a matrix of transitions $T$, with row $i$ encoding the transition vector $\phi(s_t) - \phi(s_{t-1})$, where $\phi(\cdot)$ denotes a fixed linear feature representation *known beforehand* ($i$ can be different from $t$ if transitions are observed more than once). Machado et al. (2017) justifies this sampling strategy with the fact that, in the tabular case, if every transition is sampled *once*, the right eigenvectors of matrix $T$ converge to PVFs. Because transitions are added only once, regardless of their frequency, this algorithm is not well suited to stochastic environments. In this paper we introduce an algorithm that naturally deals with stochasticity and that does not require $\phi(\cdot)$ to be known beforehand. Our algorithm learns the environment's DIF model while learning a representation of the environment from raw pixels.

## 2.3 THE SUCCESSOR REPRESENTATION

The successor representation (SR; Dayan, 1993) determines state generalization by how similar its successor states are. It is defined to be the expected future occupancy of state $s'$ given the agent's policy is $\pi$ and its starting state is $s$. It can be seen as defining state similarity in terms of time. See Figure 1 for an example. The Euclidean distance between state A and state C is smaller than the Euclidean distance between state A and state B. However, if one considers the gray tiles to be walls, an agent in state A can reach state B much *quicker* than state C. The SR captures this distinction, ensuring that state A is more similar to state B than it is to state C.

Let $\mathbb{1}_{\{\cdot\}}$ denote the indicator function, the SR, $\Psi_\pi(s, s')$, is formally defined, for $\gamma < 1$, as :

$$\Psi_\pi(s, s') \quad = \quad \mathbb{E}_{\pi, p}\left[\sum_{t=0}^\infty \gamma^t \mathbb{1}_{\{S_t = s'\}} \Big| S_0 = s\right].$$

This expectation can be estimated from samples with temporal-difference error (Sutton, 1988):

$$\hat{\Psi}(s, j) \quad \leftarrow \quad \hat{\Psi}(s, j) + \eta\left[\mathbb{1}_{\{s=j\}} + \gamma\hat{\Psi}(s', j) - \hat{\Psi}(s, j)\right], \tag{2}$$

where $\eta$ is the step-size. In the limit, the SR converges to $\Psi_\pi = (I - \gamma T_\pi)^{-1}$. This lets us decompose the value function into the product between the SR and the *immediate* reward (Dayan, 1993):

$$v_\pi(s) = \sum_{s' \in \mathcal{S}} \Psi_\pi(s, s') r(s').$$

The SR is directly related to several other ideas in the field. It can be seen as the dual approach to dynamic programming and to value-function based methods in reinforcement learning (Wang et al., 2007). Moreover, the eigenvectors generated from its eigendecomposition are equivalent to proto-value functions (Stachenfeld et al., 2014; 2017) and to slow feature analysis (Sprekeler, 2011).

| **Alg. 1** Eigenoption discovery through the SR | **Alg. 2** LEARNREPRESENTATION() with the SR |
|---|---|
| $\hat{\Psi} \leftarrow$ LEARNREPRESENTATION()
$E \leftarrow$ EXTRACTEIGENPURPOSES($\hat{\Psi}$)
**for each** eigenpurpose $\mathbf{e}_i \in E$ **do**
$\quad \langle \mathcal{I}_{e_i}, \pi_{e_i}, \mathcal{T}_{e_i} \rangle \leftarrow$ LEARNEIGENOPTION($\mathbf{e}_i$)
**end for** | **for** a given number of steps $n$ **do**
$\quad$ Observe $s \in \mathcal{S}$, take action $a \in \mathcal{A}$ selected according to $\pi(s)$, and observe a next state $s' \in \mathcal{S}$
$\quad$ **for each** state $j \in \mathcal{S}$ **do**
$\quad\quad \hat{\Psi}(s,j) \leftarrow \hat{\Psi}(s,j) +$
$\quad\quad\quad\quad \eta\big(\mathbb{1}_{\{s=j\}} + \gamma\hat{\Psi}(s',j) - \hat{\Psi}(s,j)\big)$
$\quad$ **end for**
**end for**
**return** $\hat{\Psi}$ |

Such equivalences play a central role in the algorithm we describe in the next section. The SR may also have an important role in neuroscience. Stachenfeld et al. (2014; 2017) recently suggested that the successor representation is encoded by the hippocampus, and that a low-dimensional basis set representing it is encoded by the enthorhinal cortex. Interestingly, both hippocampus and entorhinal cortex are believed to be part of the brain system responsible for spatial memory and navigation.

## 3 EIGENOPTION DISCOVERY

In order to discover eigenoptions, we first need to obtain the eigenpurposes through the eigenvectors encoding the DIF model in the environment. This is currently done through PVFs, which the agent obtains by either explicitly building the environment's adjacency matrix or by enumerating all of the environment's transitions (*c.f.* Section 2.2). Such an approach is fairly effective in deterministic settings in which states can be enumerated and uniquely identified, *i.e.*, the tabular case. However, there is no obvious extension of this approach to stochastic settings. It may be hard for the agent to explicitly model the environment dynamics in a weight matrix. The existent alternative, to enumerate the environment's transitions, may have a large cost. These issues become worse when states cannot be enumerated, *i.e.*, the function approximation case. The existing algorithm that is applicable to the function approximation setting requires a fixed representation as input, not being able to learn a representation while estimating the DIF model.

In this paper we introduce an algorithm that addresses the aforementioned issues by estimating the DIF model through the SR. Also, we introduce a new neural network that is capable of approximating the SR from raw pixels by learning a latent representation of game screens. The learned SR is then used to discover eigenoptions, replacing the need for knowing the combinatorial Laplacian. In this section we discuss the proposed algorithm in the tabular case, the equivalence between PVFs and the SR, and the algorithm capable of estimating the SR, and eigenoptions, from raw pixels.

### 3.1 THE TABULAR CASE

The general structure of the algorithms capable of discovering eigenoptions is fairly straightforward, as shown in Alg. 1. The agent learns (or is given) a representation that captures the DIF model (*e.g.*, the combinatorial Laplacian). It then uses the eigenvectors of this representation to define eigenpurposes (EXTRACTEIGENPURPOSES), the intrinsic reward functions described by Equation 1 that it will learn how to maximize. The option's policy is the one that maximizes this new reward function, while a state $s$ is defined to be terminal with respect to the eigenpurpose $\mathbf{e}_i$ if $q_*^{\mathbf{e}_i}(s,a) \leq 0$ for all $a \in \mathcal{A}$. The initiation set of an option $\mathbf{e}_i$ is defined to be $\mathcal{S} \setminus \mathcal{T}_{\mathbf{e}_i}$.

In the tabular case, our proposed algorithm is also fairly simple. Instead of assuming the matrix $\hat{\Psi}$ is given in the form of the graph Laplacian, or trying to estimate the graph Laplacian from samples by stacking the row vectors corresponding to the different observed transitions, we estimate the DIF model through the successor representation (*c.f.* Alg. 2). This idea is supported by the fact that, for our purposes, the eigenvectors of the normalized Laplacian and the eigenvectors of the SR are equivalent. Below we formalize this concept and discuss its implications. We show that the eigenvectors of the normalized Laplacian are equal to the eigenvectors of the SR scaled by $\gamma^{-1} D^{1/2}$.

The aforementioned equivalence ensures that the eigenpurposes extraction and the eigenoption learning steps remain unchanged. That is, we still obtain the eigenpurposes from the eigendecomposition[1] of matrix $\hat{\Psi}$, and we still use each eigenvector $\mathbf{e}_i \in E$ to define the new learning problem in which the agent wants to maximize the eigenpurpose, defined in Equation 1.

Importantly, the use of the SR addresses some other limitations of previous work: 1) it deals with stochasticity in the environment and in the agent's policy naturally; 2) its memory cost is independent on the number of samples drawn by the agent; and 3) it does not assume that for every action there is another action the agent can take to return to the state it was before, *i.e.*, $W$ is symmetric.

## 3.2 RELATIONSHIP BETWEEN PVFS AND THE SR

As aforementioned, PVFs (the eigenvectors of the normalized Laplacian) are equal to the eigenvectors of the successor representation scaled by $\gamma^{-1}D^{1/2}$. To the best of our knowledge, this equivalence was first explicitly discussed by Stachenfeld et al. (2014). We provide below a more formal statement of such an equivalence, for the eingevalues and the eigenvectors of both approaches. We use the proof to further discuss the extent of this interchangeability.

**Theorem.** *Stachenfeld et al. (2014): Let $0 < \gamma < 1$ s.t. $\Psi = (I - \gamma T)^{-1}$ denotes the matrix encoding the SR, and let $\mathcal{L} = D^{-1/2}(D - W)D^{-1/2}$ denote the matrix corresponding to the normalized Laplacian, both obtained under a uniform random policy. The $i$-th eigenvalue ($\lambda_{SR,i}$) of the SR and the $j$-th eigenvalue ($\lambda_{PVF,j}$) of the normalized Laplacian are related as follows:*

$$\lambda_{PVF,j} = \left[1 - (1 - \lambda_{SR,i}^{-1})\gamma^{-1}\right]$$

*The $i$-th eigenvector ($\mathbf{e}_{SR,i}$) of the SR and the $j$-th eigenvector ($\mathbf{e}_{PVF,j}$) of the normalized Laplacian, where $i + j = n + 1$, with $n$ being the total number of rows (and columns) of matrix $T$, are related as follows:*

$$\mathbf{e}_{PVF,j} = (\gamma^{-1}D^{1/2})\mathbf{e}_{SR,i}$$

*Proof.* Let $\lambda_i$, $\mathbf{e}_i$ denote the $i$-th eigenvalue and eigenvector of the SR, respectively. Using the fact that the SR is known to converge, in the limit, to $(I - \gamma T)^{-1}$ (through the Neumann series), we have:

$$
\begin{aligned}
(I - \gamma T)^{-1}\mathbf{e}_i &= \lambda_i \mathbf{e}_i \\
(I - \gamma T)\mathbf{e}_i &= \lambda_i^{-1}\mathbf{e}_i \\
(I - T)\gamma^{-1}\mathbf{e}_i &= [1 - (1 - \lambda_i^{-1})\gamma^{-1}]\gamma^{-1}\mathbf{e}_i \\
(I - T)\gamma^{-1}\mathbf{e}_i &= \lambda_j'\gamma^{-1}\mathbf{e}_i \\
(I - D^{-1}W)\gamma^{-1}\mathbf{e}_i &= \lambda_j'\gamma^{-1}\mathbf{e}_i \\
D^{-1/2}(D - W)D^{-1/2}D^{1/2}\gamma^{-1}\mathbf{e}_i &= \lambda_j'\gamma^{-1}D^{1/2}\mathbf{e}_i \qquad \square
\end{aligned}
\tag{3}
$$

Importantly, when using PVFs we are first interested in the eigenvectors with the corresponding smallest eigenvalues, as they are the "smoothest" ones. However, when using the SR we are interested in the eigenvectors with the *largest* eigenvalues. The change of variables in Eq. 3 highlights this fact *i.e.*, $\lambda_j' = [1 - (1 - \lambda_i^{-1})\gamma^{-1}]$. The indices $j$ are sorted in the reverse order of the indices $i$. This distinction can be very important when trying to estimate the relevant eigenvectors. Finding the largest eigenvalues/eigenvectors is statistically more robust to noise in estimation and does not depend on the lowest spectrum of the matrix. Moreover, notice that the scaling by $D^{1/2}$ does not change the direction of the eigenvectors when the size of the action set is constant across all states. This is often the case in the RL problems being studied.

## 3.3 THE FUNCTION APPROXIMATION CASE: THE SR THROUGH DEEP NEURAL NETWORKS

The tabular case is interesting to study because it provides intuition about the problem and it is easier to analyze, both empirically and theoretically. However, the tabular case is only realizable

---

[1]Notice the matrix $\hat{\Psi}$ is not guaranteed to be symmetric. In that case one can define the eigenpurposes to be $\hat{\Psi}$'s right eigenvectors, as we do in Section 3.3.

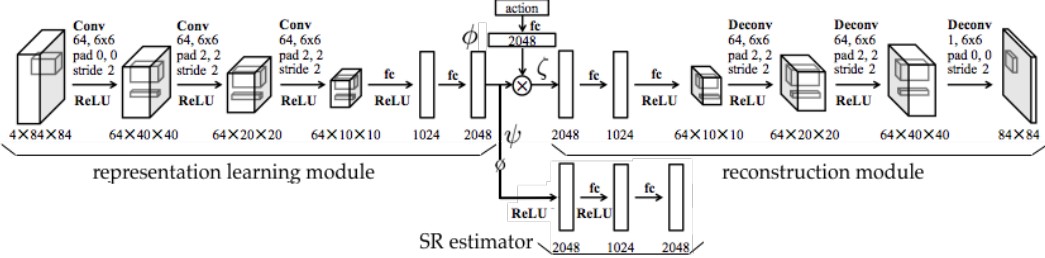

Figure 2: Neural network architecture used to learn the SR. The symbols $\otimes$ and $\emptyset$ denote element-wise multiplication and the fact that gradients are not propagated further back, respectively.

in toy domains. In real-world situations the number of states is often very large and the ability to generalize and to recognize similar states is essential. In this section, inspired by Kulkarni et al.'s (2016b) and Oh et al.'s (2015) work, we propose replacing Alg. 2 by a neural network that is able to estimate the successor representation from raw pixels. Such an approach circumvents the limitations of previous work that required a *linear* feature representation to be provided beforehand.

*The SR with non-enumerated states:* Originally, the SR was not defined in the function approximation setting, where states are described in terms of feature vectors. Successor features are the natural extension of the SR to this setting. We use Barreto et al.'s (2017) definition of successor features, where $\psi_{\pi,i}(s)$ denotes the successor feature $i$ of state $s \in \mathcal{S}$ when following a policy $\pi$:

$$\psi_{\pi,i}(s) \;=\; \mathbb{E}_{\pi,p}\left[\sum_{t=0}^{\infty}\gamma^t\phi_i(S_t)\Big|S_0 = s\right].$$

In words, $\psi_{\pi,i}(s)$ encodes the discounted expected value of the $i$-th feature in the vector $\phi(\cdot)$ when the agent starts in state $s$ and follows the policy $\pi$. The update rule presented in Eq. 2 can be naturally extended to this definition. The temporal-difference error in the update rule can be used as a differentiable loss function, allowing us to estimate the successor features with a neural network.

*Neural network architecture:* The architecture we used is depicted in Fig 2. The reconstruction module is the same as the one introduced by Oh et al. (2015), but augmented by the SR estimator (the three layers depicted at the bottom). The SR estimator uses the learned latent representation as input *i.e.*, the output of the representation learning module.

The proposed neural network receives raw pixels as input and learns to estimate the successor features of a lower-dimension representation learned by the neural network. The loss function $\mathcal{L}_{SR}$ we use to learn the successor features is:

$$\mathcal{L}_{SR}(s, s') = \mathbb{E}\left[\left(\phi^-(s) + \gamma\psi^-\left(\phi^-(s')\right) - \psi\left(\phi(s)\right)\right)^2\right],$$

where $\phi(s)$ denotes the feature vector encoding the learned representation of state $s$ and $\psi(\cdot)$ denotes the estimated successor features. In practice, $\phi(\cdot)$ is the output of the representation learning module and $\psi(\cdot)$ is the output of the SR estimator, as shown in Fig. 2. The loss function above also highlights the fact that we have two neural networks. We use $^-$ to represent a *target* network (Mnih et al., 2015), which is updated at a slower rate for stability purposes.

We cannot directly estimate the successor features from raw pixels using only $\mathcal{L}_{SR}$ because zero is one of its fixed points. This is the reason we added Oh et al.'s (2015) reconstruction module in the proposed network. It behaves as an auxiliary task (Jaderberg et al., 2017) that predicts the *next* state to be observed given the current state and action. By predicting the next state we increase the likelihood the agent will learn a representation that takes into consideration the pixels that are under its control, which has been shown to be a good bias in RL problems (Bellemare et al., 2012). Such an auxiliary task is defined through the network's reconstruction error $\mathcal{L}_{RE}$:

$$\mathcal{L}_{RE}(s, a, s') = \left(\zeta\left(\phi(s), a\right) - s'\right)^2,$$

where $\zeta(\cdot)$ denotes the output of the reconstruction module, as shown in Fig. 2. The final loss being optimized is $\mathcal{L}(s, a, s') = \mathcal{L}_{RE}(s, a, s') + \mathcal{L}_{SR}(s, s')$.

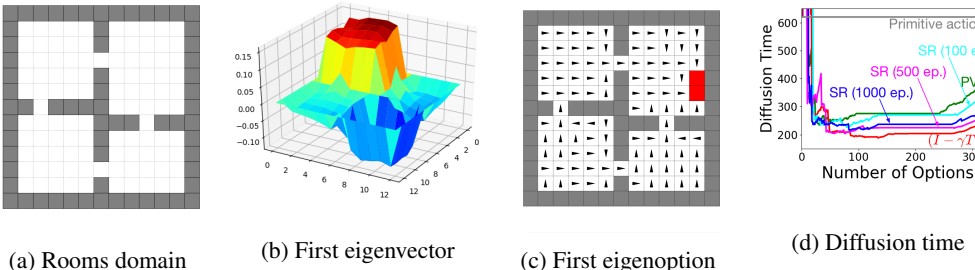

(a) Rooms domain  (b) First eigenvector  (c) First eigenoption  (d) Diffusion time

Figure 3: Results in the rooms domain. The rightmost figure depicts the diffusion time as eigenoptions are added to the agent's action set (sorted by eigenvalues corresponding to the eigenpurposes).

Finally, to ensure that the SR will not interfere with the learned features, we zero the gradients coming from the SR estimator (represented with the symbol ∅ in Fig. 2). We trained our model with RMSProp and we followed the same protocol Oh et al. (2015) used to initialize the network.

*Eigenoption learning:* In Alg. 1, the function EXTRACTEIGENPURPOSES returns the eigenpurposes described by Eq. 1. Eigenpurposes are defined in terms of a feature representation $\phi(s_t)$ of the environment and of the eigenvectors $\mathbf{e}_i$ of the DIF model (the SR in our case). We use the trained network to generate both. It is trivial to obtain $\phi(s_t)$ as we just use the output of the appropriate layer in the network as our feature representation. To obtain $\mathbf{e}_i$ we first need to generate a meaningful matrix since our network outputs a *vector* of successor features instead of a matrix. We do so by having the agent follow the uniform random policy while we store the network outputs $\psi(s_t)$, which correspond to the network estimate of the successor features of state $s_t$. We then create a matrix $T$ where row $t$ corresponds to $\psi(s_t)$ and we define $\mathbf{e}_i$ to be its right eigenvectors.

Once we have created the eigenpurposes, the option discovery problem is reduced to a regular RL problem where the agent aims to maximize the cumulative sum of rewards. Any learning algorithm can be used for that. We provide details about our approach in the next section.

## 4 EXPERIMENTS

We evaluate the discovered eigenoptions quantitatively and qualitatively in this section. We use the traditional rooms domain to evaluate the impact, on the eigenvectors and on the discovered options, of approximating the DIF model through the SR. We then use Atari 2600 games to demonstrate how the proposed network does discover purposeful options from raw pixels.

### 4.1 TABULAR CASE

Our first experiment evaluates the impact of estimating the SR from samples instead of assuming the DIF model was given in the form of the normalized Laplacian. We use the rooms domain (Fig. 3a; Sutton et al., 1999) to evaluate our method. Fig. 4b depicts the first eigenvector obtained from the SR while Fig. 4c depicts the corresponding eigenoption. We followed the uniform random policy for 1,000 episodes to learn the SR. Episodes were 100 time steps long. We used a step-size of 0.1, and we set $\gamma = 0.9$. The estimated eigenvector is fairly close to the true one and, as expected, the obtained eigenvector is fairly similar to the PVFs that are obtained for this domain. In the Appendix we provide the plots for the true SR and the PVF, as well as plots for different eigenvectors, comparing them to those obtained from $(I - \gamma T)^{-1}$.

Eigenoptions are known for improving the agent's ability to explore the environment. We use the metric diffusion time to validate whether such an ability is preserved with our method. The diffusion time can be seen as a proxy for how hard it is for an agent to reach the goal state when following a uniform random policy. It is defined as the expected number of decisions (action selection steps) an agent needs to take, when following the uniform random policy, to navigate between two randomly chosen states. We compared the agent's diffusion time when using eigenoptions obtained with PVFs to the diffusion time when using eigenoptions obtained with estimates of the SR. As we can see in Fig 3d, the eigenoptions obtained with the SR do help the agent to explore the environment. The

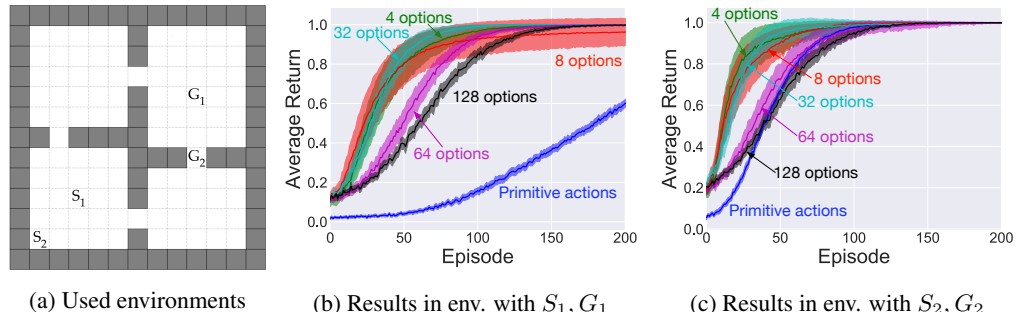

(a) Used environments          (b) Results in env. with $S_1, G_1$          (c) Results in env. with $S_2, G_2$

Figure 4: Different environments (varying start and goal locations) used in our evaluation (a), as well as the learning curves obtained in each one of these environments (b, c) for different number of options obtained from the SR when estimated after $100$ episodes. See text for more details.

gap between the diffusion time when using PVFs and when using the SR is likely due to different ways of dealing with corners. The SR implicitly models self-loops in the states adjacent to walls, since the agent takes an action and it observes it did not move.

We also evaluated how the estimates of the SR evolve as more episodes are used during learning, and its impact in the diffusion time (Fig 3d). In the Appendix we present more results, showing that the local structure of the graph is generally preserved. Naturally, more episodes allow us to learn more accurate estimates of the SR as a more global facet of the environment is seen, since the agent has more chances to further explore the state space. However, it seems that even the SR learned from few episodes allow us to discover useful eigenoptions, as depicted in Fig. 3d. The eigenoptions obtained from the SR learned using only $100$ episodes are already capable of reducing the agent's diffusion time considerably. Finally, it is important to stress that the discovered options do more than randomly selecting subgoal states. "Random options" only reduce the agent's diffusion time when hundreds of them are added to the agent's action set (Machado et al., 2017).

Finally, we evaluated the use of the discovered eigenoptions to maximize reward. In our experiments the agent learned, off-policy, the greedy policy over primitive actions (target policy) while following the uniform random policy over actions and eigenoptions (behavior policy). We used Q-learning (Watkins & Dayan, 1992) in our experiments – parameters $\lambda = 0$, $\alpha = 0.1$, and $\gamma = 0.9$. As before, episodes were $100$ time steps long. Figure 4 summarizes the obtained results comparing the performance of our approach to regular Q-learning over primitive actions. The eigenoptions were extracted from estimates of the SR obtained after $100$ episodes. The reported results are the average over $24$ independent runs when learning the SR, with each one of these runs encoding $100$ runs evaluating Q-Learning. The options were added following the sorting provided by the eigenvalues. For example, *4 options* denotes an agent with the action set used in the behavior policy being composed of the four primitive actions and the four eigenoptions generated by the top 2 eigenvalues (both directions are being used). Notice that these results do not try to take the sample efficiency of our approach into consideration, they are only meant to showcase how eigenoptions, once discovered, can speed up learning. The sample complexity of learning options is generally justified in lifelong learning settings where they are re-used over multiple tasks (*e.g.*, Brunskill & Li, 2014). This is beyond the scope of this paper.

The obtained results clearly show that eigenoptions are not only capable of reducing the diffusion time in the environment but of also improving the agent's control performance. They do so by increasing the likelihood that the agent will cover a larger part of the state space given the same amount of time. Moreover, as before, it seems that a very accurate estimate of the successor representation is not necessary for the eigenoptions to be useful. Similar results can be obtained for different locations of the start and goal states, and when the estimates of the SR are more accurate. These results can be seen in the Appendix.

## 4.2 ATARI 2600

This second set of experiments evaluates the eigenoptions discovered when the SR is obtained from raw pixels. We obtained the SR through the neural network described in Section 3. We used four

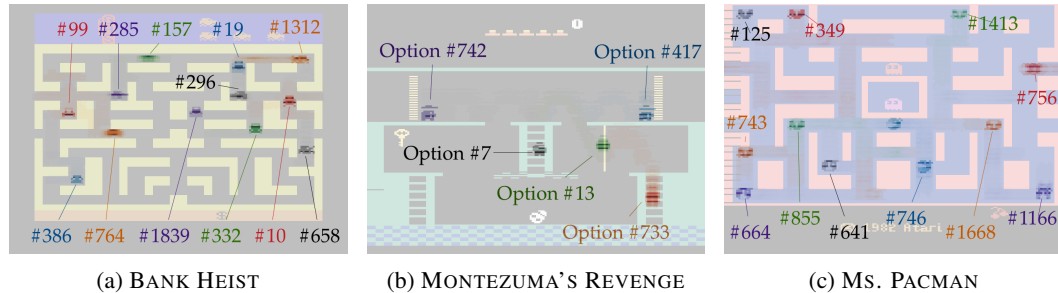

(a) BANK HEIST      (b) MONTEZUMA'S REVENGE      (c) MS. PACMAN

Figure 5: Plots of density of state visitation of eigenoptions discovered in three Atari 2600 games. States visited more frequently show darker images of the avatar. Note that an eigenoption's overwhelming mass of visitations corresponds to its terminal state, and that disparate options have different terminal states.

Atari 2600 games from the Arcade Learning Environment (Bellemare et al., 2013) as testbed: BANK HEIST, FREEWAY, MONTEZUMA'S REVENGE, and MS. PAC-MAN.

We followed the protocol described in the previous section to create eigenpurposes. We trained the network in Fig. 2 to estimate the SR under the uniform random policy. Since the network does not impact the policy being followed, we built a dataset of $500,000$ samples for each game and we used this dataset to optimize the network weights. We passed through the shuffled dataset 10 times, using RMSProp with a step size of $10^{-4}$. Once we were done with the training, we let the agent follow a uniform random policy for $50,000$ steps while we stored the SR output by the network for each observed state as a row of matrix $T$. We define $\mathbf{e}$, in the eigenpurposes we maximize (*c.f.*, Eq. 1), to be the right eigenvectors of the matrix $T$, while $\phi(\cdot)$ is extracted at each time step from the network in Fig. 2. Due to computational constraints, we approximated the final eigenoptions. We did so by using the ALE's internal emulator to do a one-step lookahead and act greedily with respect to each eigenpurpose (in practice, this is equivalent to learning with $\gamma = 0$). This is not ideal because the options we obtain are quite limited, since they do not deal with delayed rewards. However, even in such limiting setting we were able to obtain promising results, as we discuss below.

Following Machado et al. (2017), we evaluate the discovered eigenoptions qualitatively. We execute all options following the procedure described above (greedy one-step lookahead) while tracking the avatar's position on the screen. Figure 5 summarizes the behavior of some of the meaningful options discovered. The trajectories generated by different options are represented by different colors and the color's intensity at a given location represents how often the agent was at that location. Eigenoptions were introduced as options that generate purposeful behavior and that help agents explore the environment. We can clearly see that the discovered eigenoptions are indeed purposeful. They aim to reach a specific location and stay there. If this was not the case the agent's trajectory would be much more visible. Instead, what we actually observe is that the mass of visitation is concentrated on one location on the screen, dominating (color intensity) all the others. The location the agent is spending most of its time on can in fact be seen as the option's terminal state. Constantly being in a state suggests the agent has arrived to a myopic local maximum for that eigenpurpose.

In three out of four games (BANK HEIST, MONTEZUMA'S REVENGE, MS. PACMAN) our algorithm discovers options that clearly push the agent to corners and to other relevant parts of the state space, corroborating the intuition that eigenoptions also improve exploration. In MONTEZUMA'S REVENGE, the terminal state of the highlighted options even correspond to what are considered good subgoals for the game (Kulkarni et al., 2016a). It is likely that additional subgoals, such as the key, were not found due to our myopic greedy approach. This approach may also explain why our algorithm was ineffective in FREEWAY. Avoiding cars may be impossible without longer-term planning. A plot depicting the two meaningful options discovered in this game is in the Appendix. Importantly, the fact that myopic policies are able to navigate to specific locations and stay there also suggests that, as in the tabular case, the proposed approach gives rise to dense intrinsic rewards that are very informative. This is another important constrast between randomly assigned subgoals and our approach. Randomly assigned subgoals do not give rise to such dense rewards. Thus, one can argue that our approach does not only generate useful options but it also gives rise to dense eigenpurposes, making it easier to build the policies associated with them.

It is important to stress that our algorithm was able to discover eigenoptions, *from raw pixels*, similar to those obtained by algorithms that use the RAM state of the game as a feature representation. The RAM state of the game often uses specific bytes to encode important information of the game, such as the position of the player's avatar in the game. Our algorithm had to implicitly learn what were the meaningful parts of the screen. Also, different from previous algorithms, our approach is not constrained by the dimensionality of the state representation nor to binary features. Based on this discussion, we consider our results to be very promising, even though we only depict options that have effect on the initial state of the games. We believe that in a more general setting (*e.g.*, using DQN to learn policies) our algorithm has the potential to discover even better options.

## 5 RELATED WORK

Our work was directly inspired by Kulkarni et al. (2016b), the first to propose approximating the SR using a neural network. We use their loss function in a novel architecture. Because we are not directly using the SR for control, we define the SR in terms of states, instead of state-action pairs. Different from Kulkarni et al. (2016b), our network does not learn a reward model and it does not use an autoencoder to learn a representation of the world. It tries to predict the *next* state the agent will observe. The prediction module we used was introduced by Oh et al. (2015). Because it predicts the next state, it implicitly learns representations that take into consideration the parts of the screen that are under the agent's control. The ability to recognize such features is known as *contingency awareness*, and it is known to have the potential to improve agents' performance (Bellemare et al., 2012). Kulkarni et al. (2016b) did suggest the deep SR could be used to find bottleneck states, which are commonly used as subgoals for options, but such an idea was not further explored. Importantly, Jong et al. (2008) and Machado et al. (2017) have shown that options that look for bottleneck states can be quite harmful in the learning process.

The idea of explicitly building hierarchies based on the *learned* latent representation of the state space is due to Machado et al. (2017) and Vezhnevets et al. (2017). Machado et al. (2017) proposed the concept of *eigenoptions*, but limited to the linear function approximation case. Vezhnevets et al. (2017) do not explicitly build options with initiation and termination sets. Instead, they learn a hierarchy through an end-to-end learning system that does not allow us to easily retrieve options from it. Finally, Kompella et al. (2017) has proposed the use of slow feature analysis (SFA; Wiskott & Sejnowski, 2002) to discover options. Sprekeler (2011) has shown that, given a specific choice of adjacency function, PVFs (and consequently the SR) are equivalent to SFA. However, their work is limited to linear function approximation. Our method also differs in how we define the initiation and termination sets. The options they discover look for bottleneck states, which is not our case.

## 6 CONCLUSION

In this paper we introduced a new algorithm for *eigenoption* discovery in RL. Our algorithm uses the successor representation (SR) to estimate the model of diffusive information flow in the environment, leveraging the equivalence between proto-value functions (PVFs) and the SR. This approach circumvents several limitations from previous work: (i) it builds increasingly accurate estimates using a constant-cost update-rule; (ii) it naturally deals with stochastic MDPs; (iii) it does not depend on the assumption that the transition matrix is symmetric; and (iv) it does not depend on handcrafted feature representations. The first three items were achieved by simply using the SR instead of the PVFs, while the latter was achieved by using a neural network to estimate the SR.

The proposed framework opens up multiple possibilities for investigation in the future. It would be interesting to evaluate the compositionality of eigenoptions, or how transferable they are between similar environments, such as the different modes of Atari 2600 games (Machado et al., 2018). Finally, now that the fundamental algorithms have been introduced, it would be interesting to investigate whether one can use eigenoptions to accumulate rewards instead of using them for exploration.

### ACKNOWLEDGMENTS

The authors would like to thank Craig Sherstan and Martha White for feedback on an earlier draft, Kamyar Azizzadenesheli, Marc G. Bellemare and Michael Bowling for useful discussions, and the anonymous reviewers for their feedback and suggestions.

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

APPENDIX: SUPPLEMENTARY MATERIAL

This supplementary material contains details omitted from the main text due to space constraints. The list of contents is below:

- A more detailed proof of the theorem in the paper;

- Empirical results evaluating how the number of episodes used to learn the successor representation impacts the obtained eigenvectors and their corresponding eigenoptions;

- Evaluation of the reconstruction module (auxiliary task) that learns the latent representation that is used to estimate the successor representation.

A MORE DETAILED PROOF OF THE THEOREM IN THE MAIN PAPER

**Theorem.** *Stachenfeld et al. (2014): Let $0 < \gamma < 1$ s.t. $\Psi = (I - \gamma T)^{-1}$ denotes the matrix encoding the SR, and let $\mathcal{L} = D^{-1/2}(D - W)D^{-1/2}$ denote the matrix corresponding to the normalized Laplacian, both obtained under a uniform random policy. The $i$-th eigenvalue ($\lambda_{SR,i}$) of the SR and the $j$-th eigenvalue ($\lambda_{PVF,j}$) of the normalized Laplacian are related as follows:*

$$\lambda_{PVF,j} = \left[ 1 - (1 - \lambda_{SR,i}^{-1})\gamma^{-1} \right]$$

*The $i$-th eigenvector ($\mathbf{e}_{SR,i}$) of the SR and the $j$-th eigenvector ($\mathbf{e}_{PVF,j}$) of the normalized Laplacian, where $i + j = n + 1$, with $n$ being the total number of rows (and columns) of matrix $T$, are related as follows:*

$$\mathbf{e}_{PVF,j} = (\gamma^{-1}D^{1/2})\mathbf{e}_{SR,i}$$

*Proof.* This proof is more detailed than the one presented in the main paper. Let $\lambda_i$, $\mathbf{e}_i$ denote the $i$-th eigenvalue and eigenvector of the SR. Using the fact that the SR is known to converge, in the limit, to $(I - \gamma T)^{-1}$ (through the Neumann series), we have:

$$
\begin{aligned}
(I - \gamma T)^{-1}\mathbf{e}_i &= \lambda_i \mathbf{e}_i \\
(I - \gamma T)(I - \gamma T)^{-1}\mathbf{e}_i &= \lambda_i (I - \gamma T)\mathbf{e}_i \\
\mathbf{e}_i &= \lambda_i (I - \gamma T)\mathbf{e}_i \\
(I - \gamma T)\mathbf{e}_i &= \lambda_i^{-1}\mathbf{e}_i \\
(I - \gamma T)\gamma^{-1}\mathbf{e}_i &= \lambda_i^{-1}\gamma^{-1}\mathbf{e}_i \\
\gamma^{-1}\mathbf{e}_i - T\mathbf{e}_i &= \lambda_i^{-1}\gamma^{-1}\mathbf{e}_i \\
T\mathbf{e}_i &= \gamma^{-1}\mathbf{e}_i - \lambda_i^{-1}\gamma^{-1}\mathbf{e}_i \\
&= (1 - \lambda_i^{-1})\gamma^{-1}\mathbf{e}_i \\
I\mathbf{e}_i - T\mathbf{e}_i &= I\mathbf{e}_i - (1 - \lambda_i^{-1})\gamma^{-1}\mathbf{e}_i \\
(I - T)\mathbf{e}_i &= [\gamma - (1 - \lambda_i^{-1})]\gamma^{-1}\mathbf{e}_i \\
(I - T)\gamma^{-1}\mathbf{e}_i &= [1 - (1 - \lambda_i^{-1})\gamma^{-1}]\gamma^{-1}\mathbf{e}_i
\end{aligned}
$$

$$
\begin{aligned}
(I - T)\gamma^{-1}\mathbf{e}_i &= \lambda_j'\gamma^{-1}\mathbf{e}_i \\
(I - D^{-1}W)\gamma^{-1}\mathbf{e}_i &= \lambda_j'\gamma^{-1}\mathbf{e}_i \\
(D^{-1}(D - W))\gamma^{-1}\mathbf{e}_i &= \lambda_j'\gamma^{-1}\mathbf{e}_i \\
D^{1/2}(D^{-1}(D - W))\gamma^{-1}\mathbf{e}_i &= \lambda_j'\gamma^{-1}D^{1/2}\mathbf{e}_i \\
D^{-1/2}(D - W)\gamma^{-1}\mathbf{e}_i &= \lambda_j'\gamma^{-1}D^{1/2}\mathbf{e}_i \\
D^{-1/2}(D - W)D^{-1/2}D^{1/2}\gamma^{-1}\mathbf{e}_i &= \lambda_j'\gamma^{-1}D^{1/2}\mathbf{e}_i \\
\mathcal{L}D^{1/2}\gamma^{-1}\mathbf{e}_i &= \lambda_j'\gamma^{-1}D^{1/2}\mathbf{e}_i \qquad \square
\end{aligned}
$$

THE IMPACT THE NUMBER OF EPISODES HAS IN LEARNING THE SR AND THE EIGENOPTIONS

In Section 4.1 we briefly discussed the impact of estimating the successor representation from samples instead of assuming the agent has access to the normalized Laplacian. It makes much more sense to use the successor representation as the DIF model in the environment if we can estimate it quickly. The diffusion time was the main evidence we used in Section 4.1 to support our claim that early estimates of the successor representation are useful for eigenoption discovery. In order to be concise we did not actually plot the eigenvectors of the estimates of the successor representation at different moments, nor explicitly compared them to proto-value functions or to the eigenvectors of the matrix $(I - \gamma T)^{-1}$. We do so in this section.

Figures 7–10 depict the first four eigenvectors of the successor representation in the Rooms domain, after being learned for different number of episodes (episodes were 100 time steps long, $\eta = 0.1$, $\gamma = 0.9$). We also depict the corresponding eigenvectors of the $(I - \gamma T)^{-1}$ matrix[2], and of the normalized Laplacian (Machado et al., 2017). Because the eigenvectors orientation (sign) is often arbitrary in an eigendecomposition, we matched their orientation to ease visualization.

Overall, after 500 episodes we already have an almost perfect estimate of the first eigenvectors in the environment; while 100 episodes seem to not be enough to accurately learn the DIF model in all rooms. However, learning the successor representation for 100 episodes seems to be enough to generate eigenoptions that reduce the agent's diffusion time, as we show in Figure 3d. We can better discuss this behavior by looking at Figures 11–14, which depict the options generated by the obtained eigenvectors.

With the exception of the options generated after learning the successor representation for 100 episodes, all the eigenoptions obtained from estimates of the successor representation already move the agent towards the "correct" room(s). Naturally, they do not always hit the corners, but the general structure of the policies can be clearly seen. We also observe that the eigenoptions obtained from proto-value functions are shifted one tile from the corners. As discussed in the main paper, this is a consequence of how Machado et al.'s (2017) dealt with corners. They did not model self-loops in the MDP, despite the fact that the agent can be in the same state for two consecutive steps. The successor representation captures this naturally. Finally, we use Figure 11a to speculate why the options learned after 100 episodes are capable of reducing the agent's diffusion time. The first eigenoption learned by the agent moves it to the parts of the state space it has never been to, this may be the reason that the combination of these options is so effective. It also suggests that incremental methods for option discovery and exploration are a promising path for future work.

USING EIGENOPTIONS TO ACCUMULATE REWARD IN THE ENVIRONMENT

In Section 4.1 we also evaluated the agent's ability to accumulate reward after the eigenoptions have been learned. We further analyze this topic here. As in Section 4.1, the agent learned, off-policy, the greedy policy over primitive actions (target policy) while following the uniform random policy over actions and eigenoptions (behavior policy). We used Q-learning (Watkins & Dayan, 1992) in our experiments – parameters $\lambda = 0$, $\alpha = 0.1$, and $\gamma = 0.9$. Episodes were 100 time steps long. Figures 16–19 summarize the obtained results comparing the performance of our approach to regular Q-learning over primitive actions in four different environments (*c.f.* Figure 15). We evaluate the agent's performance when using eigenoptions extracted from estimates of the SR obtained after 100, 500, and 1000 episodes, as well eigenoptions obtained from the true SR, *i.e.*, $(I - \gamma T)^{-1}$. The reported results are the average over 24 independent runs when learning the SR, with each one of these runs encoding 100 runs evaluating Q-Learning. The options were added following the sorting provided by the eigenvalues. For example, *4 options* denotes an agent with the action set used in the behavior policy being composed of the four primitive actions and the four eigenoptions generated by the top 2 eigenvalues (both directions are being used).

We can see that eigenoptions are not only capable of reducing the diffusion time in the environment but of also improving the agent's control performance. They do so by increasing the likelihood that the agent will cover a larger part of the state space given the same amount of time. Interestingly, few eigenoptions seem to be enough for the agent. Moreover, although rough estimates of the SR seem to be enough to improve the agent's performance (*e.g.*, estimates obtained after only 100 episodes).

---

[2]Recall $(I - \gamma T)^{-1}$ is the matrix to which the successor representation converges to in the limit.

More accurate predictions of the SR are able to further improve the agent's performance, mainly when dozens of eigenoptions are being used. The first eigenoptions to be accurately estimated are those with larger eigenvalues, which are the ones we add first.

### EVALUATION OF THE RECONSTRUCTION TASK

In Section 4.2 we analyzed the eigenoptions we are able to discover in four games of the Arcade Learning Environment. We did not discuss the performance of the proposed network in the auxiliary tasks we defined. We do so here. Figures 20–23 depict a comparison between the target screen that should be predicted and the network's actual prediction for ten time steps in each game. We can see that it accurately predicts the general structure of the environment and it is able to keep track of most moving sprites on the screen. The prediction is quite noisy, different from Oh et al.'s (2015) result. Still, it is interesting to see how even an underperforming network is able to learn useful representations for our algorithm. It is likely better representations would result in better options.

### EIGENOPTIONS DISCOVERED IN FREEWAY

Figure 6 depicts the two meaningful eigenoptions we were able to discover in the game FREEWAY. As in Figure 5, each option is represented by the normalized count of the avatar's position on the screen in a trajectory. The trajectories generated by different options are represented by different colors and the color's intensity at a given location represents how often the agent was at that location.

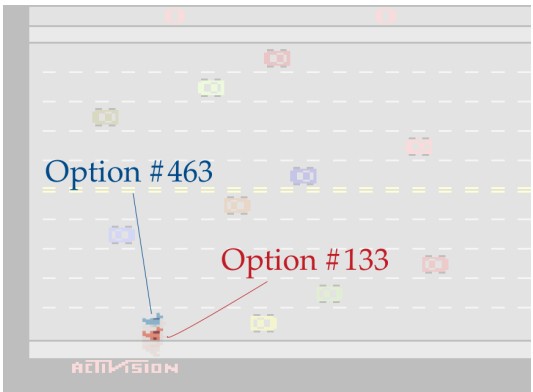

Figure 6: Eigenoptions discovered in the game FREEWAY.

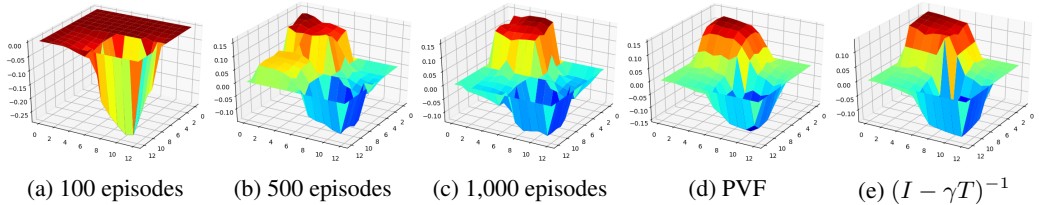

(a) 100 episodes     (b) 500 episodes     (c) 1,000 episodes     (d) PVF     (e) $(I - \gamma T)^{-1}$

Figure 7: Evolution of the **first eigenvector** being estimated by the SR and baselines.

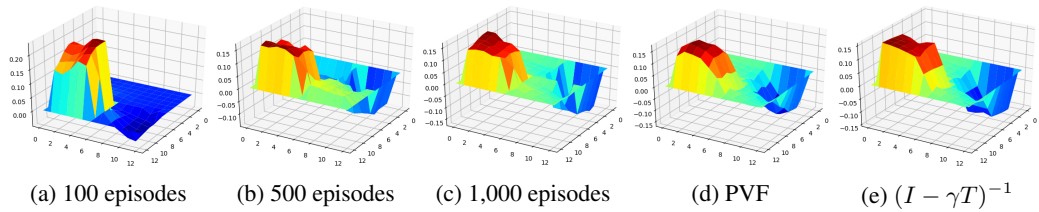

(a) 100 episodes     (b) 500 episodes     (c) 1,000 episodes     (d) PVF     (e) $(I - \gamma T)^{-1}$

Figure 8: Evolution of the **second eigenvector** being estimated by the SR and baselines.

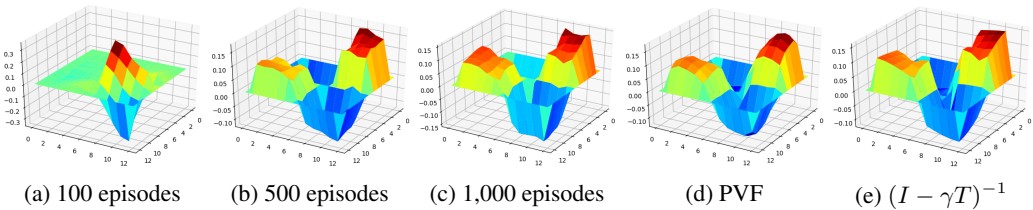

(a) 100 episodes     (b) 500 episodes     (c) 1,000 episodes     (d) PVF     (e) $(I - \gamma T)^{-1}$

Figure 9: Evolution of the **third eigenvector** being estimated by the SR and baselines.

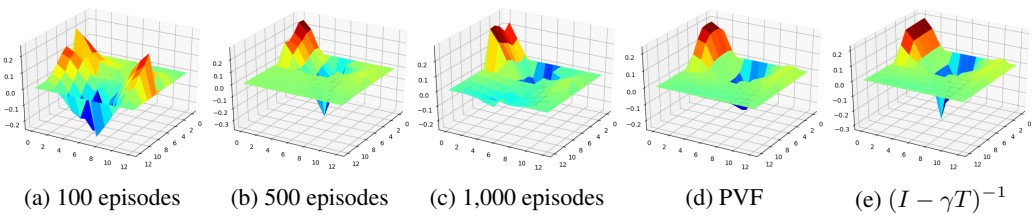

(a) 100 episodes     (b) 500 episodes     (c) 1,000 episodes     (d) PVF     (e) $(I - \gamma T)^{-1}$

Figure 10: Evolution of the **fourth eigenvector** being estimated by the SR and baselines.

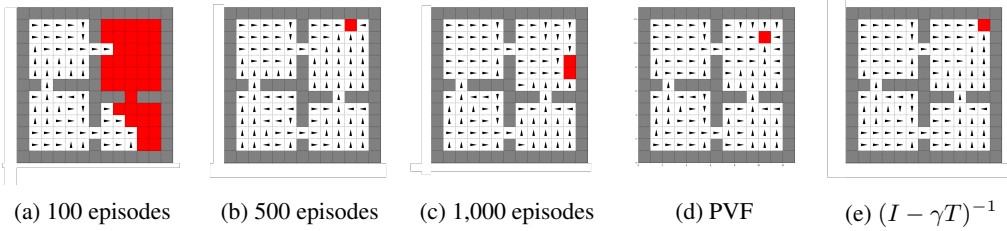

(a) 100 episodes    (b) 500 episodes    (c) 1,000 episodes    (d) PVF    (e) $(I - \gamma T)^{-1}$

Figure 11: Evolution of the **first eigenoption** being estimated by the SR and baselines.

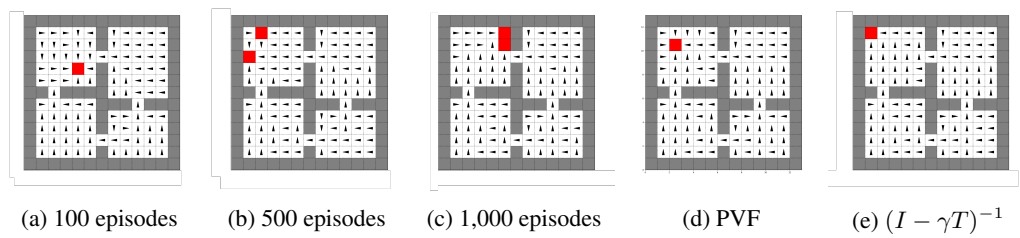

(a) 100 episodes    (b) 500 episodes    (c) 1,000 episodes    (d) PVF    (e) $(I - \gamma T)^{-1}$

Figure 12: Evolution of the **second eigenoption** being estimated by the SR and baselines.

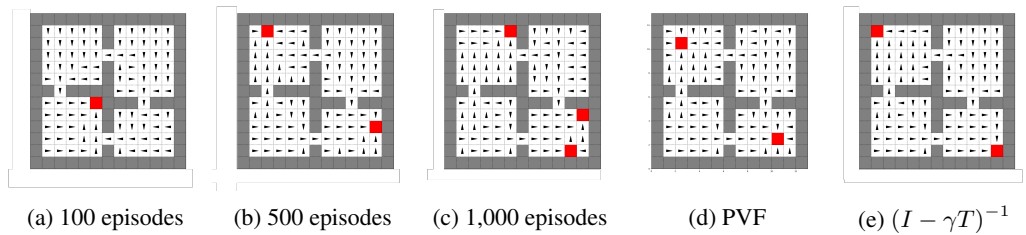

(a) 100 episodes    (b) 500 episodes    (c) 1,000 episodes    (d) PVF    (e) $(I - \gamma T)^{-1}$

Figure 13: Evolution of the **third eigenoption** being estimated by the SR and baselines.

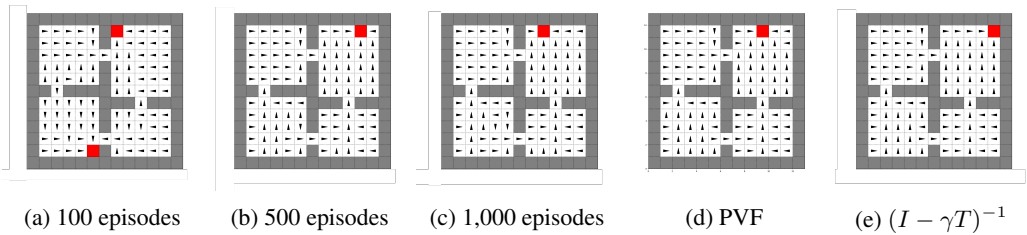

(a) 100 episodes    (b) 500 episodes    (c) 1,000 episodes    (d) PVF    (e) $(I - \gamma T)^{-1}$

Figure 14: Evolution of the **fourth eigenoption** being estimated by the SR and baselines.

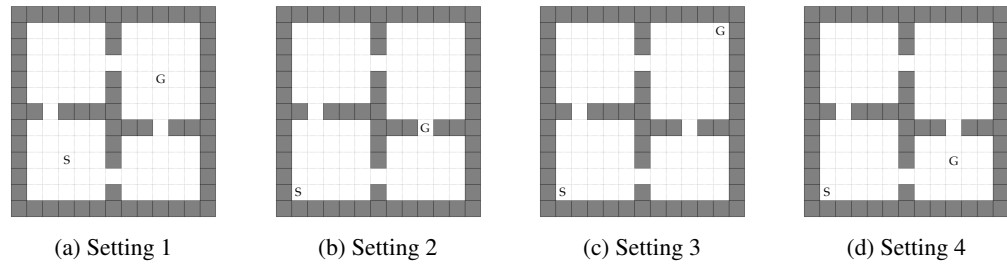

(a) Setting 1      (b) Setting 2      (c) Setting 3      (d) Setting 4

Figure 15: Different environments (varying start and goal locations) used when evaluating the agent's ability to accumulate reward with and without eigenoptions.

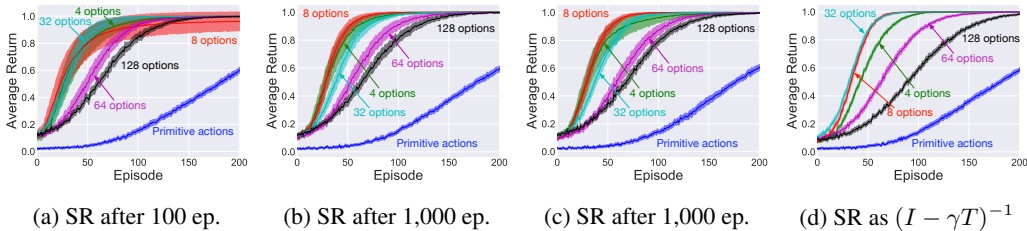

(a) SR after 100 ep.    (b) SR after 1,000 ep.    (c) SR after 1,000 ep.    (d) SR as $(I - \gamma T)^{-1}$

Figure 16: Plot depicting the agent's performance when following options obtained through estimates of the SR (100, 500, and 1,000 episodes), as well as through the true SR, in environment 1.

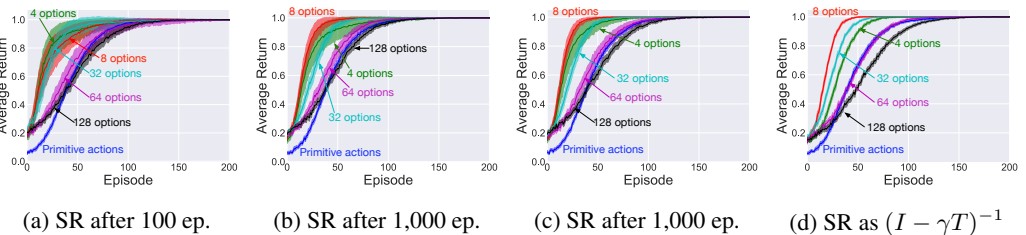

(a) SR after 100 ep.    (b) SR after 1,000 ep.    (c) SR after 1,000 ep.    (d) SR as $(I - \gamma T)^{-1}$

Figure 17: Plot depicting the agent's performance when following options obtained through estimates of the SR (100, 500, and 1,000 episodes), as well as through the true SR, in environment 2.

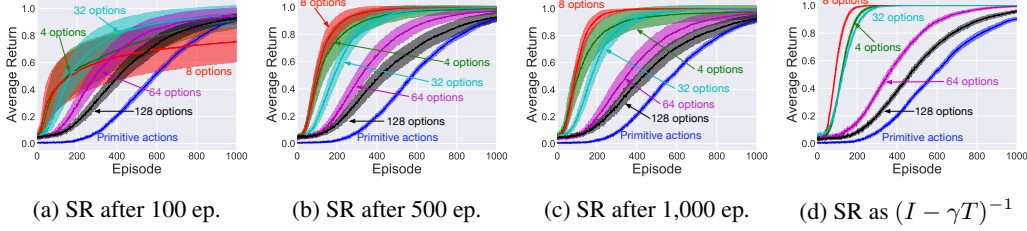

(a) SR after 100 ep.    (b) SR after 500 ep.    (c) SR after 1,000 ep.    (d) SR as $(I - \gamma T)^{-1}$

Figure 18: Plot depicting the agent's performance when following options obtained through estimates of the SR (100, 500, and 1,000 episodes), as well as through the true SR, in environment 3.

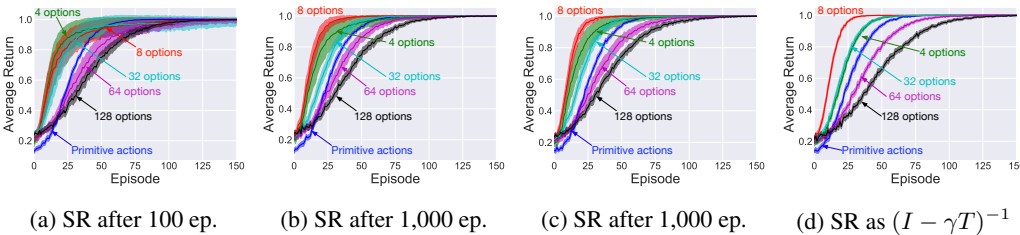

(a) SR after 100 ep.    (b) SR after 1,000 ep.    (c) SR after 1,000 ep.    (d) SR as $(I - \gamma T)^{-1}$

Figure 19: Plot depicting the agent's performance when following options obtained through estimates of the SR (100, 500, and 1,000 episodes), as well as through the true SR, in environment 4.

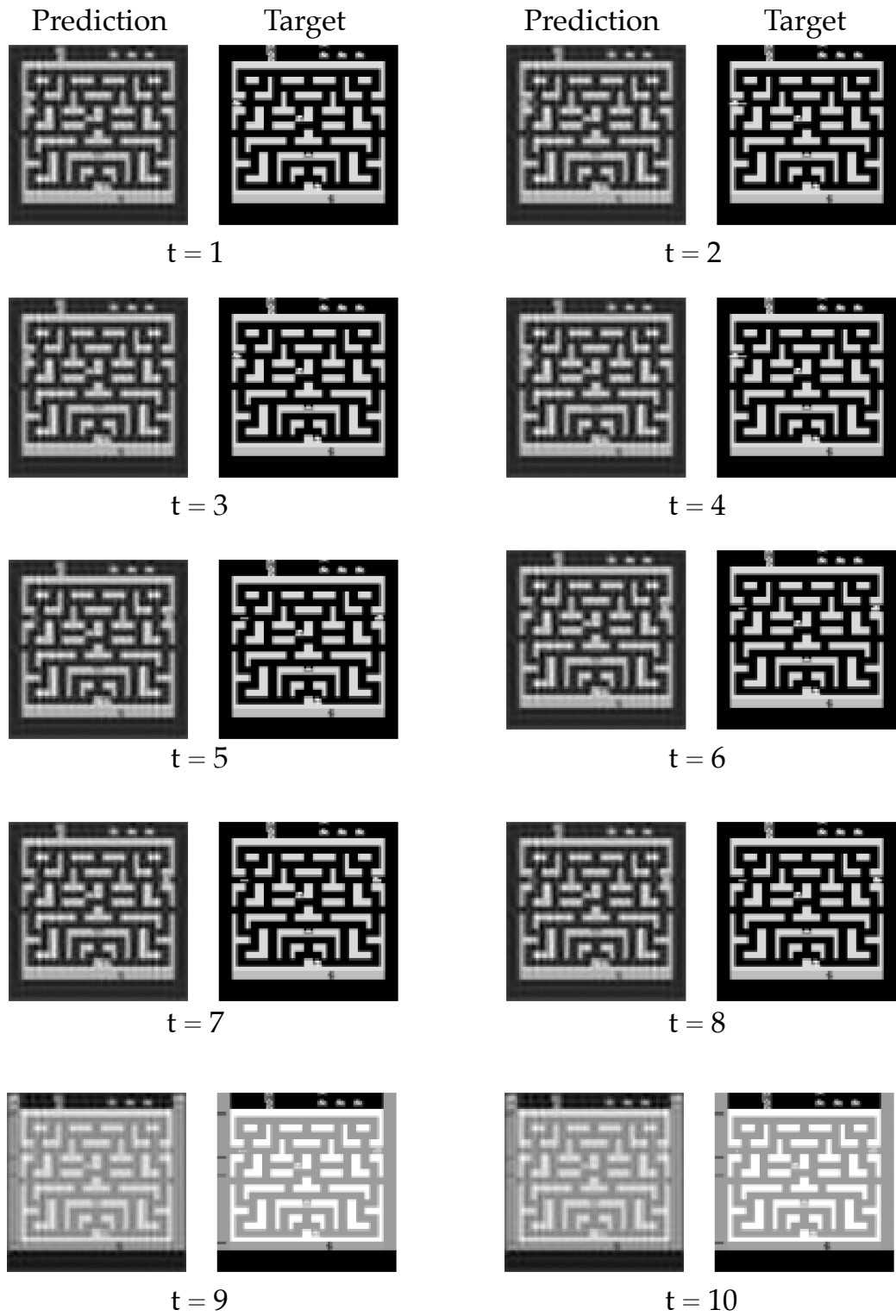

Figure 20: Final 1-step predictions in the game BANK HEIST. We use the task of predicting the next game screen as an auxiliary task when estimating the successor representation.

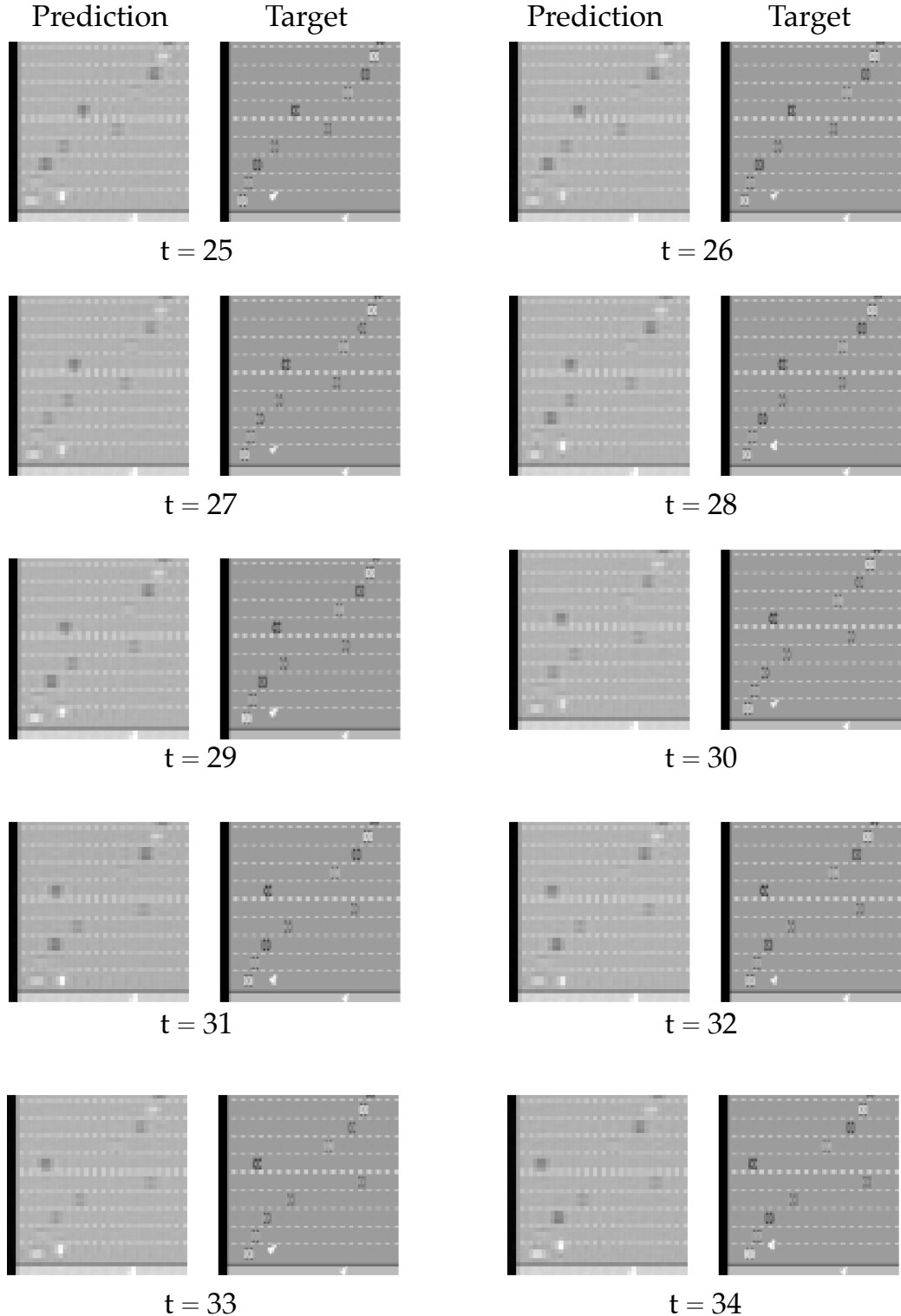

Figure 21: Final 1-step predictions in the game FREEWAY. We use the task of predicting the next game screen as an auxiliary task when estimating the successor representation.

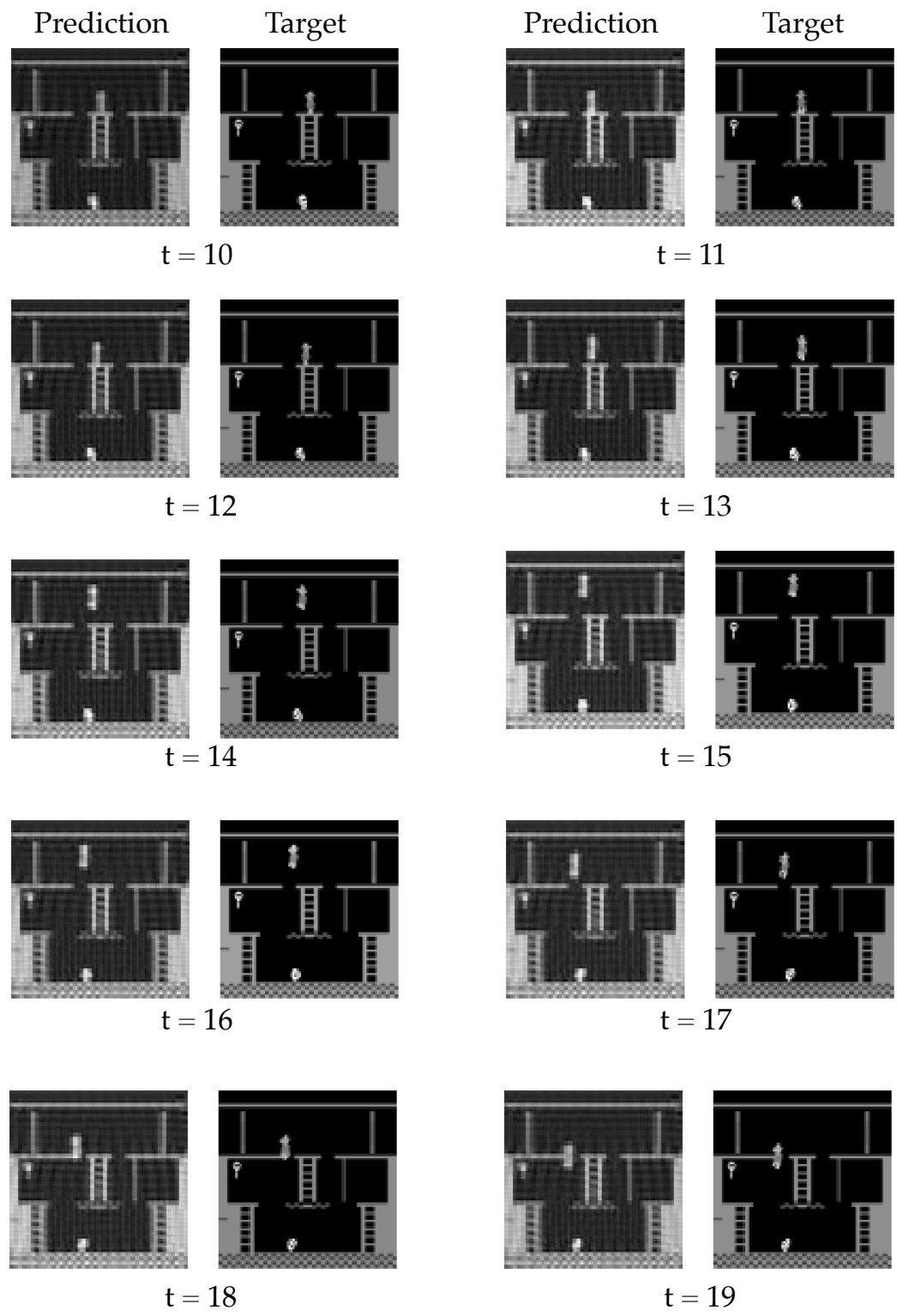

Figure 22: Final 1-step predictions in the game MONTEZUMA'S REVENGE. We use the task of predicting the next game screen as an auxiliary task when estimating the successor representation.

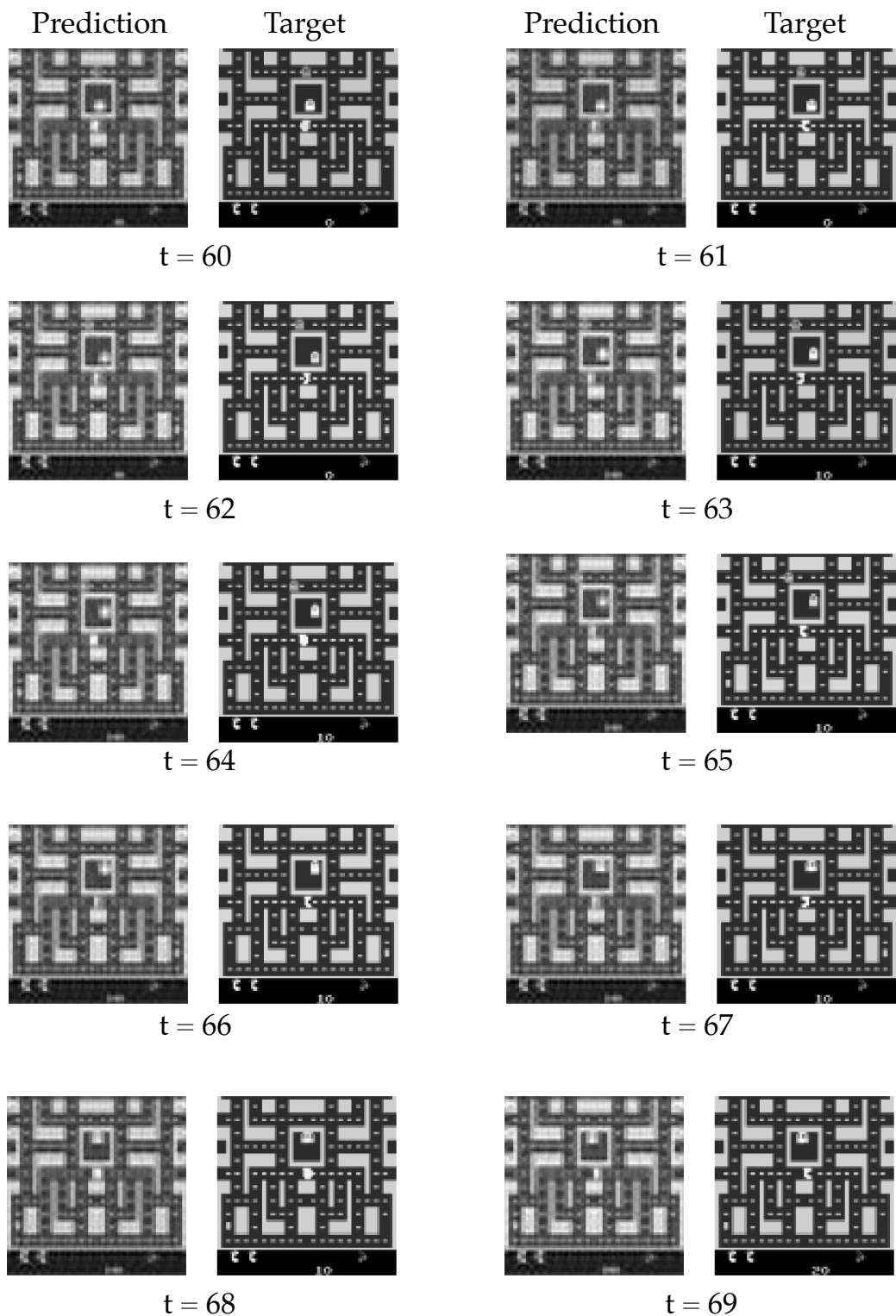

Figure 23: Final 1-step predictions in the game MS. PACMAN. We use the task of predicting the next game screen as an auxiliary task when estimating the successor representation.

