# OpenReview forum: "Eigenoption Discovery through the Deep Successor Representation"
_ICLR.cc/2018/Conference — Accept (Poster)_

### Official Review · AnonReviewer2 · 2017-11-26
**Interesting preliminary extension of eigenoptions to stochastic domains with learned non-linear features**

**Rating:** 6
**Confidence:** 3

**Review:**

The paper extends the idea of eigenoptions, recently proposed by Machado et al. to domains with stochastic transitions and where state features are learned. An eigenoption is defined as an optimal policy for a reward function defined by an eigenvector of the matrix of successor representation (SR), which is an occupancy measure induced here by a uniform policy. In high-dimensional state space, the authors propose to approximate that matrix with a convolutional neural network (CNN). The approach is evaluated in a tabular domain (i.e., rooms) and Atari games.

Overall the paper is well-written and quite clear. The proposed ideas for the extension seem natural (i.e., use of SR and CNN). The theorem stated in the paper seems to provide an interesting link between SR and the Laplacian. However, a few points are not clear to me:
- Is the result new or not? If I understand correctly, Stachenfeld et al. discussed this result, but didn't prove it. Is that correct? So the provided proof is new?
- Besides, how are D and W exactly defined?
- Finally, as the matrix is not symmetric, do real eigenvalues always exist?

The execution of the proposed ideas in the experiments was a bit disappointing to me. The approximated eigenoption was simply computed as a one-step greedy policy. Besides, the eigenoptions seem to help for exploration (as a uniform policy was used) as indicated by plot 3(d), but could they help for other tasks (e.g., learn to play Atari games faster or better)? I think that would be a more useful measure for the learned eigenoptions.

During learning SR and the features, what would be the impact if the gradient for SR estimation were also propagated?

In Figure 4, the trajectories generated by the different eigenoptions are barely visible.

Some typos:
- Section 2.1:
in the definition of G_t, the expectation is taken over p as well
I_w and T_w should be a subset of S

- in (2), the hat is missing over \Psi
in the definition of v_\pi(s), r only depends on s'? This seems inconsistent with the previous definition of \psi

- p. 6:
in the definition of L_{SR}(s, s'), why \psi takes \phi(s) as argument?

- in conclusion:
that that

---

> ### Author Response · Authors · 2017-12-13
> **Thank you for the comments. We improved the execution in the proposed ideas in the experiments with a new set of results. Also, we improved the paper write-up after reading your questions, which are responded below.**
>
> Thank you for your feedback and thorough review. We believe our paper is better now that we took your input into consideration.
>
> Regarding the execution of the proposed ideas in the experiments, in the new version of our submission we provide a different measure of the usefulness of eigenoptions. We now also show, in the tabular case, how they can be used to improve the agent’s control performance (Figure 4 in the main text and Figures 16-19 in the Appendix). In this new set of experiments the agent takes a random walk over eigenoptions while learning to maximize reward with primitive actions. Such an approach speeds up learning dramatically as a consequence of the agent being able to better explore the environment.
>
> The responses to the other questions asked are itemized below:
>
> -- Is the result new or not? If I understand correctly, Stachenfeld et al. discussed this result, but didn't prove it. Is that correct? So the provided proof is new?
>
> Yes, that is correct. Stachenfeld et al. (2014) discussed the result but did not provide a formal proof of it, nor the relationship between the eigenvalues of both approaches. Also, because the authors provided an informal discussion, they were not very precise in their claims, ignoring for example the fact that the equivalence we discuss is true only if the generated graph is regular (i.e., the size of the action set is the same across every state). To be precise, below is what Stanchenfeld et al. (2014) wrote: “Under a random walk policy, the transition matrix is given by $T = D^{-1}W$. If $\phi$ is an eigenvector of the random walk’s graph Laplacian $I - T$, then $D^{1/2} \phi$ is an eigenvector of the normalized graph Laplacian. The corresponding eigenvector for the discounted Laplacian, $I-\gamma T$, is $\gamma \phi$. Since the matrix inverse preserves the eigenvectors, the normalized graph Laplacian has the same eigenvectors as the SR, $M = (I - \gamma T)^{-1}$, scaled by $\gamma D^{-1/2}$.”
>
>
> -- Besides, how are D and W exactly defined?
>
> D and W are defined for PVFs. We use the same definition given by Machado et al. (2017): W is the graph’s adjacency matrix and D is the diagonal matrix whose entries are the row sums of W. W(i, j) is defined to be 1 if there is an action that allows the agent to go from state i to state j.
>
> -- As the matrix is not symmetric, do real eigenvalues always exist?
>
> We thank the reviewer for the question about the matrix symmetry and its eigenvalues because we made this discussion clearer in the paper now. The eigenvalues/eigenvector are not necessarily real for the eigendecomposition of an asymmetric matrix. However, the right eigenvectors obtained from the singular value decomposition of the matrix are always real (this is what we do in the ALE experiments).
>
> -- During learning SR and the features, what would be the impact if the gradient for SR estimation were also propagated?
>
> We did not investigated this possibility. Because this is the first demonstration of eigenoptions discovery from raw pixels, we wanted to keep the learning process as simple as possible. Thus, we avoided the interaction between the loss function of the SR and the reconstruction error. This is something we plan to investigate in future work.
>
> -- In Figure 4, the trajectories generated by the different eigenoptions are barely visible.
>
> This was in fact intentional. We did not want to use those results to focus on the trajectory that led the agent to the highlighted state, but on the final state itself. We did so by plotting the mass of visitation of each state. If the trajectories were visible, it would mean that the agent was navigating through the environment without a clear purpose. We wanted to show the exact opposite. That the agent was clearly spending the vast majority of the time in a specific location. We made this clearer in the updated version of our submission.
>
> -- in the definition of L_{SR}(s, s'), why \psi takes \phi(s) as argument?
>
> In the definition of $L_{SR}(s, s’)$, $\psi$ takes $\phi(s)$ as argument because we are implicitly referring to Figure 2, in which we labeled the output of some layers as functions. We define $\psi$ to be the SR module while $\phi$ is the output of the representation learning module. We really appreciate this question, it made us realize the need to further clarify this in the paper, which we also did in the updated version of our submission.
>
> Naturally, we also fixed all the typos you listed (thank you for that).

---

### Official Review · AnonReviewer3 · 2017-11-28

**Rating:** 9
**Confidence:** 5

**Review:**

- This paper shows an equivalence between proto value functions and successor representations. It then derives the idea of eigen options from the successor representation as a mechanism for option discovery. The paper shows that even under a random policy, the eigen options can lead to purposeful options

- I think this is an important conceptual paper. Automatic option discovery from raw sensors is perhaps one of the biggest open problems in RL research. This paper offers a new conceptual setup to look at the problem and consolidates different views (successor repr, proto values, eigen decomposition) in a principled manner.

- I would be keen to see eigen options being used inside of the agent. Have authors performed any experiments ?

- How robust are the eigen options for the Atari experiments? Basically how hand picked were the options?

- Is it possible to compute eigenoptions online? This seems crucial for scaling up this approach

---

> ### Author Response · Authors · 2017-12-13
> **Thank you for the comments. We introduced new experiments showing eigenoptions being used inside of the agent. We answer your questions below.**
>
> Thank you for your kind review. We have just updated our submission to include a new set of results where the eigenoptions are used inside of the agent. We show how eigenoptions can also be used to improve the agent’s control performance (Figure 4 in the main text and Figures 16-19 in the Appendix). We show that one can take a random walk over eigenoptions while learning to maximize reward with primitive actions. Such an approach speeds up learning dramatically, since the agent can better explore the environment.
>
> Regarding the robustness of the eigenoptions in the Atari experiments, we were able to select the options in a fairly straightforward way: by looking at those that generated a high density of visitation in a particular location on the screen. We did have multiple similar options we ended-up not reporting for clarity. We also had options that would not move the agent to anywhere (probably because of gamma being set to 0) and others in which the agent was happy regardless of the action taken (likely because the agent was trying to maximize features that were not under its control). We consider the results presented in this paper promising because we were able to replicate, using raw pixels, the results Machado et al. (2017) obtained when using the RAM state of the game (that encodes explicit information the agent cares about). However, we do think some extra work still needs to be done on option pruning. We do have some ideas, such as pruning options based on whether the agent can in fact maximize the returned generated by the eigenpurpose and pruning options that lead to the same distribution that other options lead. This is something we want to further investigate in a future work to allow us to easily obtain a set of useful options.
>
> Finally, we are very excited about the direction of research you asked about (computing eigenoptions online). This is definitely something we are planning to investigate in the future work. It should be possible to compute the eigenoptions online. There are incremental methods capable of estimating the singular value decomposition of a matrix, aside from  other methods capable of discovering the top k eigenvectors of a matrix (notice that our method is much more stable than eigenoptions obtained from PVFs, which needs to estimate the *bottom* k eigenvectors). Once we have the eigenvectors, we could actually learn all eigenoptions simultaneously through off-policy learning. Also, it is not far fetched to imagine an algorithm that learns the intra-option policy and the policy over options simultaneously, bootstrapping from the option-critic architecture, for instance.

---

### Official Review · AnonReviewer1 · 2017-12-01
**Good paper connecting previous works from the literature to propose an algorithm for automatic option discovery.**

**Rating:** 7
**Confidence:** 4

**Review:**

Eigenoption Discovery Through the Deep Successor Representation

The paper is a follow up on previous work by Machado et al. (2017) showing how proto-value functions (PVFs) can be used to define options called “eigenoptions”. In essence, Machado et al. (2017) showed that, in the tabular case, if you interpret the difference between PVFs as pseudo-rewards you end up with useful options. They also showed how to extend this idea to the linear case: one replaces the Laplacian normally used to build PVFs with a matrix formed by sampling differences phi(s') - phi(s), where phi are features. The authors of the current submission extend the approach above in two ways: they show how to deal with stochastic dynamics and how to replace a linear model with a nonlinear one. Interestingly, the way they do so is through the successor representation (SR). Stachenfeld et al. (2014) have showed that PVFs can be obtained as a linear transformation of the eigenvectors of the matrix formed by stacking all SRs of an MDP. Thus, if we have the SR matrix we can replace the Laplacian mentioned above. This provides benefits already in the tabular case, since SRs naturally extend to domains with stochastic dynamics. On top of that, one can apply a trick similar to the one used in the linear case --that is,  construct the matrix representing the diffusion model by simply stacking samples of the SRs. Thus, if we can learn the SRs, we can extend the proposed approach to the nonlinear case. The authors propose to do so by having a deep neural network similar to Kulkarni et al. (2016)'s Deep Successor Representation. The main difference is that, instead of using an auto-encoder, they learn features phi(s) such that the next state s' can be recovered from it (they argue that this way psi(s) will retain information about aspects of the environment the agent has control over).

This is a well-written paper with interesting (and potentially useful) insights. I only have a few comments regarding some aspects of the paper that could perhaps be improved, such as the way eigenoptions are evaluated.

One question left open by the paper is the strategy used to collect data in order to compute the diffusion model (and thus the options). In order to populate the matrix that will eventually give rise to the PVFs the agent must collect transitions. The way the authors propose to do it is to have the agent follow a random policy. So, in order to have options that lead to more direct, "purposeful" behaviour, the agent must first wander around in a random, purposeless, way, and hope that this will lead to a reasonable exploration of the state space.

This problem is not specific to the proposed approach, though: in fact, any method to build options will have to resolve the same issue. One related point that is perhaps more specific to this particular work is the strategy used to evaluate the options built: the diffusion time, or the expected number of steps between any two states of an MDP when following a random walk. First, although this metric makes intuitive sense, it is unclear to me how much it reflects control performance, which is what we ultimately care about. Perhaps more important, measuring performance using the same policy used to build the options (the random policy) seems somewhat unsatisfactory to me. To see why, suppose that the options were constructed based on data collected by a non-random policy that only visits a subspace of the state space. In this case it seems likely that the decrease in the diffusion time would not be as apparent as in the experiments of the paper. Conversely, if the diffusion time were measured under another policy, it also seems likely that options built with a random policy would not perform so well (assuming that the state space is reasonably large to make an exhaustive exploration infeasible). More generally, we want options built under a given policy to reduce the diffusion time of other policies (preferably ones that lead to good control performance).

Another point associated with the evaluation of the proposed approach is the method used to qualitatively assess options in the Atari experiments described in Section 4.2. In the last paragraph of page 7 the authors mention that eigenoptions are more effective in reducing the diffusion time than “random options” built based on randomly selected sub-goals. However, looking at Figure 4, the terminal states of the eigenoptions look a bit like randomly-selected  sub-goals. This is especially true when we note that only a subset of the options are shown: given enough random options, it should be possible to select a subset of them that are reasonably spread across the state space as well.

Interestingly, one aspect of the proposed approach that seems to indeed be an improvement over random options is made visible by a strategy used by the authors to circumvent computational constraints. As explained in the second paragraph of page 8, instead of learning policies to maximize the pseudo-rewards associated with eigenoptions the authors used a myopic policy that only looks one step ahead (which is the same as having a policy learned with a discount factor of zero). The fact that these myopic policies are able to navigate to specific locations and stay there suggests that the proposed approach gives rise to dense pseudo-rewards that are very informative. As a comparison, when we define a random sub-goal the resulting reward is a very sparse signal that would almost certainly not give rise to useful myopic policies. Therefore, one could argue that the proposed approach not only generate useful options, it also gives rise to dense pseudo-rewards that make it easier to build the policies associated with them.

---

> ### Author Response · Authors · 2017-12-13
> **Thank you for the comments. We introduced new experiments and discussions trying to address your concerns.**
>
> Thank you for such a careful analysis of our paper. The main point you raised was about how we evaluated the eigenoptions. Initially we did not evaluate the options beyond the diffusion time because this metric seems to be related to the agent’s control performance (Machado et al., 2017). However, after reading the reviews, we do realize this is not something we should gloss over. Thus, we have just updated our submission to include a new set of results, in the tabular case, showing how eigenoptions can also be used to improve the agent’s control performance (Figure 4 in the main text and Figures 16-19 in the Appendix). We show that one can take a random walk over eigenoptions while learning to maximize reward with primitive actions. Such an approach speeds up learning dramatically, since the agent can better explore the environment.
>
> We hope this new experiment also addresses, at least partially, the concern about looking at the diffusion time only over a uniform random walk. We focus on the diffusion time under a random walk because we are interested in the setting in which the agent cannot easily stumble upon a non-zero reward in the environment. In this case, most model-free agents just act randomly. We do agree that ideally we should be able to do better than the random wandering our agents do. However, this is a very hard thing to do given the fact that the agent has no information about the world, as the reviewer points out with the fact that all papers in the literature rely on that. Hopefully this paper is a step towards this direction. Our evaluation does show that augmenting the agent’s action set with eigenoptions makes the exploration process much more efficient in this case, after the short period action without purpose.
>
> We also really appreciate the reviewer’s interpretation of our results in the ALE. We added the discussion/contrast  between the sparseness of rewards generated by random subgoals and the subgoals generated by the eigenoptions. We fully agree with that point. Finally, notice it is not straightforward to define a valid random subgoal state when using function approximation because states cannot be uniquely identified. If we define, for example, a specific pixel configuration to be the random sub-goal, it is not clear we can actually observe such random configuration. Our algorithm naturally deals with this issue as well.

---

### Decision · Program_Chairs · 2018-01-29
**ICLR 2018 Conference Acceptance Decision**

**Decision:**

Accept (Poster)

**Comment:**

This paper on automatic option discovery connects recent research on successor representations with eigenoptions. This is a solidly presented, conceptual paper with results in tabular and atari environments.